# Immune Response Modulation by HPV16 Oncoproteins in Lung Cancer: Insights from Clinical and In Vitro Investigations

**DOI:** 10.3390/v16111731

**Published:** 2024-11-04

**Authors:** Bianca de França São Marcos, Daffany Luana dos Santos, Georon Ferreira de Sousa, Leonardo Carvalho de Oliveira Cruz, Bárbara Rafaela da Silva Barros, Matheus Gardini Amâncio Marques de Sena, Vanessa Emanuelle Pereira Santos, Talita Helena de Araújo Oliveira, Cristiane Moutinho Lagos de Melo, Antonio Carlos de Freitas

**Affiliations:** 1Laboratory of Molecular Studies and Experimental Therapy, Department of Genetics, Federal University of Pernambuco, Av. Prof. Moraes Rego, 1235, Cidade Universitária Recife, Recife 50670-901, PE, Brazil; bianca.saomarcos@ufpe.br (B.d.F.S.M.); daffany.luana@ufpe.br (D.L.d.S.); matheus.gardini@ufpe.br (M.G.A.M.d.S.); vanessa.emanuelle@ufpe.br (V.E.P.S.); 2Keizo Asami Immunopathology Laboratory, Federal University of Pernambuco, Av. Prof. Moraes Rego, 1235, Cidade Universitária Recife, Recife 50670-901, PE, Brazil; georon.sousa@gmail.com (G.F.d.S.); leonardo.oliveiracruz@ufpe.br (L.C.d.O.C.); barbara.sbarros@ufpe.br (B.R.d.S.B.); 3Immunological and Antitumor Analysis Laboratory, Department of Antibiotics, Federal University of Pernambuco, Recife 50670-901, PE, Brazil; 4Patrick G Jonhston Centre for Cancer Research, Queen’s University Belfast, Belfast BT7 1NN, UK; t.oliveira@qub.ac.uk

**Keywords:** lung cancer, HPV oncogenes, immune cell interaction, cancer cell signaling, macrophages and lymphocytes

## Abstract

Lung cancer has the highest mortality rates worldwide, and Human Papillomavirus (HPV) has been associated with its carcinogenesis. In this study, HPV16 genes’ expressions were investigated in patient samples, along with the immunological response promoted by lymphocytes and monocytes in A549 cells transfected with HPV oncogenes and co-cultured with PBMC. An increase in the expression of E5 was observed in the patients’ samples. In the in vitro analysis, a decrease in the number of monocytes and cytotoxic cells was observed when co-stimulated by E6 and E7, and it promoted an increase in the Th2 profile. In contrast, the high proliferation of cytotoxic cells in A549 cells transfected with E5, associated with the high expression of costimulatory molecules in monocytes, suggests a low capacity of E5 to inhibit the presentation of antigens by antigen-presenting cells (APC) and a possible use of E5 in future therapeutic strategies against lung cancers associated with HPV.

## 1. Introduction

HPV infections are the main causes of malignant and benign lesions in the cervix. In the past several years, such infections have been associated with various non-anogenital tumors, including lung cancer [1,2,3]. Among the most detected high-risk HPV types in lung tumors, HPV16 and HPV18 were the most prevalent types [3,4,5]. However, their incidence varied according to the geographical location of the study and its methodology.

HPV16 was the most prevalent in the most recent meta-analysis studies, with a high incidence in Asia, followed by Latin America, Europe, and North America [1,2]. The same was observed in a previous study published by our group, which showed that in Brazil, HPV activity was detected in lung tumor samples, with an incidence of 52%, with HPV16 being the most prevalent [6].

Worldwide lung cancer has a high mortality index with different causes such as smoking and viral infections [7]. HPV can promote cancer due to three oncoproteins (E5, E6, and E7) which act by inhibiting several tumor suppressor genes, as well as activating some transcription factors associated with uncontrolled cell proliferation in several genital, head, and neck cancers [7,8]. The first HPV association with lung cancer was published in 1970 by Syjanen et al. Since then, some studies have shown the presence of the virus in lung tumors, especially in lung adenocarcinoma [1,9,10,11,12].

HPV interacts with the tumor microenvironment, manipulating immunological cells to escape the immunological defense. The tumor microenvironment is dynamic and interactive and is formed by cancer cells, stroma cells, and immunological cells such as macrophages and recruited lymphocytes [12,13,14,15,16]. 

Although some studies have suggested a potential link between lung cancer and HPV [1,2,17], this assertion remains uncertain. Therefore, our aim here was to investigate the activity of HPV16 genes in lung tumors from patients, as well as to assess the immune response of monocytes and lymphocytes against the A549 cell line (lung adenocarcinoma) transfected with HPV16 oncogenes (E5, E6, and E7) individually in vitro. In this regard, our two main inquiries were whether there is activity in the viral proteins in patient samples and whether each of the different HPV oncogenes could induce changes in the signaling of immune cells and cancer cells, and if so, what was the behavior of this signaling?

## 2. Materials and Methods

### 2.1. Processing Patients’ Biological Material

This study included 18 biological samples obtained from patients undergoing thoracic surgery at the pulmonology department of Oswaldo Cruz University Hospital, located in Recife, Pernambuco, Brazil. The fresh/frozen biopsies were obtained and stored in sterile tubes with RNAlater solution at −80 °C. Informed consent was obtained from all patients, and the present study was approved by the Ethics Committee of the Federal University of Pernambuco, Health Sciences Center/UFPE (process number: 06396812.0.3001.5192).

All HPV16-positive biological samples were selected based on the quality of RNA extraction. The RNA was extracted from all collected tumors using Trizol reagent, and subsequently the RNeasy Mini kit (Qiagen, Hilden, Germany) was employed to isolate and purify the RNA. The extracted RNA was assessed for its integrity and quality by visualizing ribosomal RNA bands on 1% agarose gel electrophoresis and on the nanodrop (Thermo Scientific, Waltham, MA, USA), considering the 260/280 ratio. The cDNA synthesis was performed following the instructions of the Maxima First Strand cDNA Synthesis kit with dsDNase (Thermo Scientific, Waltham, MA, USA).

### 2.2. Detection HPV Activity 

HPV detection and genotyping, as well as the expression of E5, E6, and E7, were previously performed and published by our research group [18]. In the present study, we evaluate the mRNA expression of the E2 gene and compare it with the expression of HPV16 E5, E6, and E7, which were previously analyzed in the aforementioned study, in biological samples from lung cancer patients. A reanalysis of the oncoproteins was conducted and compared with previous analyses to validate the results of E2. The primers for HPV oncogenes were designed using Primer3Plus (Appendix A). The reference genes ACTB and EEF1A1, used for relative quantification, were previously validated in lung tissue [18]. Thus, the geometric mean of the reference genes was used to calculate the relative expression of all targets in the samples [19]. All samples were tested in duplicate, using the HPV16-positive cell line (C3) as a positive control for the RT-qPCR reactions. Samples with Cq values of 35 or higher were considered negative. The 2ΔΔCq method was employed to determine the expression levels of specific genes (E2, E5, E6, and E7). This calculation was based on two internal control genes (ACTB and EEF1A1) and utilized HPV16 cervical cancer samples to assess relative expression.

The gene expression analysis was performed for all 18 fresh/frozen lung tumor samples using Fast SYBR Green Master Mix (Applied Biosystems^®^, Foster city, CA, USA). The cycling conditions were 95 °C for 20 s for polymerase activation, followed by 40 cycles of 95 °C for 3 s for denaturation and 60 °C for 30 s for annealing and extension.

### 2.3. Cellular Lineage and Transfection Protocol

In this study, we utilized the A549 tumor cell line, which originates from lung adenocarcinoma. That cell lineage was cultured in DMEM (Dulbecco’s Modified Eagles Medium Invitrogen^®^, Carlsbad, CA, USA) medium supplemented with 10% of FBS (Gibico^®^-Thermo Fisher Scientific^®^-Waltham, MA, USA); L-Glutamina 1% (Sigma^®^-St. Louis, MI, USA) and antibiotics (1% penicillin and streptomycin) for a 24 h prior co-culture with lymphocytes and monocytes. The HPV16 E5, E6, and E7 oncogenes were based on a standard sequence from GenBank (K02718.1). The oncogenes were cloned into an expression vector to mammalian cells, pcDNA 3.1(+). The clones were confirmed by restriction analysis and submitted to the automatic sequencing using the sequencing kit ABI PRISM BigDyeTM terminator v3.1 (Applied Biosystems^®^). After cloning confirmation, the DNA recombinant vectors were isolated by Plus Maxi (Qiagen^®^) kit in accordance with manufacturer instructions. The transfections were performed in A549 cells, in plates of 48 wells (10^5^ cells/well), using the Lipofectamine 3000 solution (Thermo Fisher Scientific^®^-Waltham, MA, USA). Each tumor cell was transfected with 250 ng/µL of plasmid with each oncogene by well. All assays were performed in five experimental replicates.

### 2.4. Confirming the Expression of the Oncogenes 

After the transfection of the plasmid containing the HPV16 oncoproteins in the lung cell line, RNA extraction was performed following the PureLink ^®^ RNA mini Kit (Ambion-Thermo Fisher Scientific, Austin, TX, USA) protocol. The RNA underwent DNase treatment utilizing DnaseI, Rnase-free (Thermo Fisher Scientific®, Waltham, MA, USA), followed by reverse transcription using the High Capacity cDNA Reverse Transcription Kits (Applied Biosystems^®^, Foster city, CA, USA). All reactions were performed in a thermocycler-LineGene9660 (Bioer, Hangzhou, Zhejiang Province, China)-using Fast SYBR ^®^ Green Master Mix (Applied Biosystems^®^—Foster city, CA, USA).

### 2.5. Isolation of Peripheric Blood Mononuclear Cells (PBMCs) from Human Volunteers

About 8–10 mL of peripheric blood was collected in EDTA tubes of five healthy volunteer donors. The PBMCs were isolated from blood using a 1.077 g/mL Ficoll (300× *g*/30 min/26 °C) (GE healthcare^®^, Chicago, IL, USA). After two washes with PBS (300× *g*/10 min/26 °C), cells were resuspended in a supplemented DMEM medium and cultured for 24 h in 75 cm^2^ flasks (5% CO_2_/37 °C). The next day, the lymphocytes in suspension were centrifugated twice in PBS (300× *g*/30 min/26 °C), and, using the cell counter Countess 3 (Thermo Fisher^®^, Waltham, MA, USA), about 10^5^ cells were seeded in plates of 48 wells. Monocytes adherent in the same flask were removed using 2% trypsin, followed by centrifugation in PBS (300× *g*/30 min/26 °C). Equally, 10^5^ cells were seeded in plates of 48 wells. 

### 2.6. Experimental Groups and Co-Culture of A549 with Immunological Cells

To investigate the immune tumor response in vitro, A549 (10^5^ cells/well) were seeded. After 80% of the confluence (about 24 h), lymphocytes or monocytes were inserted in those cultures following the scheme shown in Figure 1, performed in quintuplicate. After 24 h of culturing, the cells were stained by immunophenotyping, and the culture supernatants were collected to evaluate the cytokine production.

### 2.7. PBMC Immunophenotyping and Cytokine Analysis

The immunophenotyping staining was performed by surface antibodies anti-CD3 (FITC or APC), -CD4 (FITC or APC), -CD8 (FITC or PE), -CD56 (Pecy5.5 or APC), -CD25 (PercP), -CD14 (FITC), -B7.1 (PE), B7.2 (APC), and HLA-DR (PercP). The intracellular antibodies anti-FoxP3 (PE), -Perforina (FITC), granzyme (PercP), -IL-17 (PE), -IFN (APC), and -IL-10 (PE) were also investigated. The A549 cells were labeled with the CD90 antibody for gating strategy (all antibodies were obtained from BD^®^). The IL-2, IL-4, IL-6, IL-10, TNF-α, IFN-γ, and IL-17 supernatant cytokines were measured by CBA kit (BD^®^). All assays were performed through a flow cytometer (Accuri BD^®^, Ann Arbor, MI, USA) in 30,000 events for cellular investigation and 2100 events for cytokine investigation.

### 2.8. Statistical Analysis

Data distribution was verified using the Kolmogorov–Smirnov test. All samples that adopted a normal distribution were used for this purpose, and a One-way ANOVA test was performed. When the samples did not follow a normal distribution, the Kruskal–Wallis test was applied. A *p*-value < 0.05 was considered statistically significant. All statistical analyses were performed using GraphPad Prism software version 9.0.0 (GraphPad Software, Inc. San Diego, CA, USA). 

## 3. Results

### 3.1. Characterization of the Biological Samples

Most of the evaluated samples were from male (66.7%), aged between 56 and 81 years (66.7%), smokers (77.8%), and alcohol consumers (61.1%) (Appendix A). It was found that HPV16 genes were expressed in the samples, suggesting a potential viral activity in lung tumor cells infected with HPV.

### 3.2. Expression of HPV Oncogenes in Patient Samples

When evaluating the expression of HPV16 genes in lung tumor samples from patients, distinct differences were observed in the detection of HPV16 genes. The E5 gene was detected in most lung tumors (9 out of 18). The second most expressed gene was the oncogene E7, detected in 4 out of 18 tumors. The oncogene E6 showed low expression compared to the others, observed in 2 out of 18 samples (data obtained from the previous study) [20]. The E2 mRNA was also weakly expressed, detected in only two samples. There was no statistically significant difference in mRNA expression levels (Appendix A).

### 3.3. CD14+ Monocytes Suffered Inhibition by A549 Tumor Cells Transfected with E6 and E7 Oncogenes

The expression of CD14+ and its co-stimulatory molecules was evaluated on the surface of monocytes co-cultured with A549 cells transfected with the empty vector, which was compared to cells transfected with the oncogenes. The E6 and E7 oncogenes inhibited monocyte activation, evidenced by a decrease in the number of monocytes (Figure 2A) and a reduction in the numbers of cells marked, particularly B7.1 and HLA-DR (Figure 2B,D).

### 3.4. A549-Transfected Cells Induced a Decrease in the Cytotoxic Action in Both CD8+ and CD56+ Lymphocytes

We evaluated the profile of CD8+ and CD56+ cells, comparing the group of A549 cells transfected with the empty vector to the group of A549 cells transfected with the oncogenes. Additionally, the intracellular production of granzyme and perforin was assessed in the different transfection groups. The numbers of CD8+ and CD56+ T cells decreased when cultured with A549 tumor cells transfected with E6 and E7, except in the E5 groups (Figure 3A,B), although no statistically significant differences were found. Regarding the cytotoxic activity of these cells, the results showed an increase in the number of CD8+ T cells producing intracellular perforin and a decrease in the number producing intracellular granzyme in the E6 and E7 groups (Figure 3C). Furthermore, there was an increase in the number of CD56+ cells with intracellular perforin and a decrease in the number with intracellular granzyme, particularly in the E6 and E7 transfected groups (Figure 3D).

### 3.5. A549-Transfected Cells Induced a Decrease in the Th1 Profile in T CD4+ Lymphocytes and Inhibited the Inflammatory Phenotype in CD56+ Lymphocytes

We evaluated the intracellular production of the cytokines IL-10, IL-17, and IFN in CD4+, CD8+, and CD56+ lymphocytes, as well as the expression profile of Treg cells (CD4+, CD25+, and FOXP3). These expressions were compared in A549 cells transfected with the empty vector and those transfected with the oncogenes, both of which were co-cultured with lymphocytes. 

CD4+ T lymphocytes were stimulated by the A549 tumor cell lineage in all experimental groups. However, especially in the transfected cell groups, those values were statistically increased. In the investigation of the CD4+ T phenotype, results showed a decrease in Th1 subsets due to lower values of IFN-γ and IL-10 intracellular cytokines for transfected A549 cells with E6 and E7 (Figure 4A,B). However, another inflammatory way, such as Th17, was activated by HPV E6 (Figure 4C) and a possible presence of the T regulatory phenotype (TregFOXP3+) was also activated by HPV oncogenes (E6 and E7) but without statistical values (Figure 4D).

All cells were co-cultured with lymphocytes. It was possible to observe a difference in stimulus to CD8+ and CD56+ lymphocytes cultured with A549 cells that were transfected. Both CD8+ and CD56+ cells did not suffer a change in the cytokine production against A549 cells, but when challenged against transfected A549 cells, different behaviors could be observed. There was no significant statistical difference in the production of IL-10 and IL-17 in transfected CD8+ lymphocytes (Figure 4F,G). However, IFN-γ showed a reduction compared to the transfected groups. CD56+ cells suffered a decrease in IFN-γ production when transfected with E6 and E7. While there was an increase in IL-10 production when co-cultured with A549/E6 and A549/E7 (Figure 4J), IL-17 did not show a difference (Figure 4I).

### 3.6. Cytokines Produced in the Supernatant of Transfected A549 Cells Co-Cultured with Lymphocytes

We compared the cytokines released into the supernatant by T lymphocytes cultured with A549 cells transfected with an empty vector to those cultured with tumor cells expressing HPV16 oncoproteins. The results demonstrated that lymphocytes cultured with oncogenes-transfected tumor cells promoted an increase in the release of the cytokines IL-2, IL-4, IL-10, and TNF-α, with differences in the releases from different transfected oncoproteins (Figure 5A,B,D,E). However, there was no statistically significant difference in the production of IL-6, IFN, and IL-17 in the supernatant of co-cultured A549-transfected cells with lymphocytes (Figure 5C,F,G). 

### 3.7. Cytokines Produced in the Supernatant of Transfected A549 Cells Co-Cultured with Monocytes

We evaluated the cytokines released in the supernatant of A549 cells transfected with the empty vector compared to those transfected with the oncogenes when co-cultured with monocytes. The results of the monocyte co-cultured with A549 lineage transfected with HPV16 oncogenes showed a significant increase in the release of cytokines, mainly IL-2, IL-6, and TNF-α (Figure 6A,C,E). However, there was no statistically significant difference in IL-4, IFN-γ, and IL-17 production between the groups evaluated (Figure 6B,F,G). The release of cytokine IL-10 was reduced in the presence of E6 in lung cells co-cultured with monocytes (Figure 6D).

## 4. Discussion

In this study, our results showed that samples from lung cancer patients exhibited HPV activity, characterized by an increased expression of E5 and E7 but low expression of E6 and E2. Moreover, our results in vitro suggest the viral oncoproteins produced by lung tumor cells modulated the activity of lymphocytes and monocytes. Our study sheds light on the activity of the E5 oncoprotein, revealing increased expression in patient samples. Additionally, contrary to expectations, E5 did not promote pro-tumor activity in infected lung adenocarcinoma cell lines. Instead, it maintained the expression of surface molecules on monocytes responsible for antigen presentation, which is different from the E6 and E7 oncoproteins.

Furthermore, we investigated the expression of the main HPV16 oncogenes (E2, E5, E6, and E7) associated with the viral infection cycle and carcinogenesis. In the lung cancer context, the expression of at least one of the HPV oncogenes was detected in 72% of the lung cancer samples (Appendix A). However, there was no statistically significant difference between the presence and absence of HPV, nor the risk of HPV infection in relation to age, sex, tobacco, and alcohol consumption, likely due to the small number of samples obtained in this study.

The HPV16 E7 and E5 oncogenes were the most expressed viral genes in our samples, unlike E6 and E2, which showed low amounts of expression in the patients’ tumor samples. Since 1990, it has been suggested that HPV E5 may cooperate with E7, similar to BPV infections, where the E5 gene can synergistically interact with E7. Together, they may contribute to the establishment and persistence of the infection, as well as the control of cell proliferation [21]. In HPV infections, it has also been observed that E5 and E7 can work together to stimulate the epidermal growth factor (EGF), increasing cell proliferation and viral DNA synthesis [22]. 

To date, only one study conducted by our group evaluated the expression of E5 in HPV-positive lung cancer samples, which found that the oncogene E5 was the most prevalent and highly expressed among the samples [20]. Nonetheless, further evaluation would be important because the E5 oncoprotein can enhance EGFR signaling through direct and indirect mechanisms in cervical cancer [23]. EGFR mutations, which are common in certain types of lung cancer, may be associated with E5 activity; these mutations were more prevalent in lung adenocarcinomas harboring HPV DNA [24,25,26,27]. Furthermore, the E5 oncogene is known to modulate various immune mechanisms, including antigen presentation and inflammatory pathways [28].

Immune cell infiltration suggests that the tumor microenvironment contributes to lung carcinogenesis and that these tumors are largely composed of T and B lymphocytes, as well as macrophages, Natural Killer (NK) cells, and dendritic cells [29]. In our study, we evaluated the activity of T CD4+, Treg, T CD8+, and T NK lymphocytes (CD56+), as well as the activity of monocytes (CD14+) and the production of cytokines by these cells when cultured together with the cell line A549 with and without the presence of HPV16 oncoproteins. Our results showed that lung tumor cells expressing HPV oncogenes could modulate the activity of these immune cells.

Macrophages and other APCs express both MHC I and II molecules, along with co-stimulatory proteins such as CD80 (B7.1) and CD86 (B7.2). These components can regulate T cell activation [30]. In 60% of head and neck tumors, the surface receptor HLA-DR (MHC II) is downregulated or lost as a strategy to evade immune recognition [31]. Moreover, cervical cancers associated with HPV suffer an augmentation of the antigen presentation, through HLA-DR, under IFN-γ treatment [32]. Here, results showed a decrease in monocyte and immunological co-stimulation by E6 and E7 HPV genes (Figure 2A–D). However, Sato et al. (2022) also showed an increase in B7 costimulatory molecules in non-small cell lung cancer. The authors suggest that this increase is associated with the worst prognosis due to CTLA-4 and/or PD1-L signalization on lung cancers. Although, their study did not evaluate the relationship between lung cancer and HPV [30]. Controversially, a study affirms that the cervical epithelium with HPV presence causes a decrease in B7.2 expression, as observed in our in vitro experiments [33].

Macrophage infiltration (M1 and M2) in and around the tumor is one of the main features of tumor progression [34]. M1 macrophages are tumor-resistant and can be identified by different phenotypic markers, including CD80/B7-1, CD86/B7-2, HLA-DR, and NOS2 [35,36]. Cancer cells can evade the host’s immune system by overexpressing B7 family molecules (such as CD86 or PD-L1), which repress T-cell antitumor responses by binding to the PD-1 receptor [37]. Interaction between CD28 and the B7 ligands CD80 (B7-1) and CD86 (B7-2) increases T cell activation and cytokine release [37]. The lower levels of IL-10 cytokines, as well as the increase in TNF (Figure 6) found in our study, suggest the M1 activation and M2 inhibition in cells transfected with HPV oncogenes, suggesting higher pro-inflammatory activity and lower anti-inflammatory activity, which does not favor tumor development. The increased activity of M2 macrophages presents a tumor-promoting capacity involving immunosuppression, angiogenesis activation, and stromal remodeling [38].

Currently, the CD4+ T lymphocytes have been studied with more attention in cancer immunotherapy due to their ability to promote CD8+ T activation and act directly or indirectly against tumor cells [33]. Here, our results showed that the HPV oncogenes were able to increase those cell numbers in the same way as observed in the CD4+ T cell infiltration in cervical cancer [39]. Although low values of intracellular IL-10 were found in CD4+ T cells (Figure 4B), the high values of that cytokine observed in the culture supernatant associated with high IL-4 (Figure 5B) suggest an anti-inflammatory profile (Th2) promotion by HPV oncogenes present in the transfected A549 tumor cells. IL-4 and IL-10 cytokines are upregulated in premalignant and cancer lesions [40,41]. IL-4 and IL-13, as well as IL-10, have been demonstrated to play important roles during primary tumor progression in mouse models of cancer [42]. Moreover, IL-10 inhibits antitumoral cytotoxic T-cell responses and hampers immune surveillance due to the blocking of antigen presentations by APCs (through MHC II reduction), and the synthesis of cytokines of the Th1 profile [43,44,45,46].

CD8+ and CD56+ lymphocytes can infiltrate and identify tumor cells, and their presence in HPV-positive tumor microenvironments is correlated with a better prognosis [36,37,38]. Our results showed that the cytotoxic cells analyzed suffered a decrease when co-cultured with A549 tumor cells transfected with HPV E6 and E7 oncogenes (Figure 3), but not with the E5 oncogene. Interestingly, the high proliferation of the CD8+ and CD56+ cells in the E5 group was also associated with the high expression of costimulatory molecules (especially HLA-DR) in monocytes (Figure 2B–D), suggesting a low ability of this oncogene to inhibit the antigen presentation by APCs. On the other hand, in cervical cancer, HPV16 E5 is associated with a decrease in the MHC (HLA) Class I complex being retained in the Golgi apparatus, preventing its transport to the surface [47].

Related to the cytotoxic action promoted by CD8+ and CD56+ lymphocytes, results showed that an accumulation of perforin in both cells (Figure 3C,D), the low production of granzyme in CD8+ cells (Figure 3C), and the low amount of intracellular IFN-γ suggest the suppression of effector action by those cells, especially by E6 and E7 oncogenes (Figure 4E). The suppression profile observed in the CD56+ cells is also reinforced by the low intracellular IFN-γ production by those cells associated with high IL-10 production in cells transfected with E6 and E7 (Figure 4H,I).

Senju et al. show that tumor cells express HPV oncoproteins and that these soluble proteins affect NK cells by reducing IFN-γ production because E6 and E7 bind to the IL-18 receptor. IL-18 signaling promotes the activation and expansion of NK cells and improves their cytotoxicity and tumor activity, as well as the CD80, CD86, and HLA-DR expression in APCs [48]. 

The cytokines evaluated here show a suppression of the Th2 immune response and a possible increase in the Th1 profile. For both lymphocytes and monocytes culture supernatants, the same behavior was found, i.e., HPV E5-, E6-, and E7-transfected cells promoted the high release of IL-2, TNF-α, and IL-6 (Figure 5 and Figure 6). Furthermore, the IL-4 and IL-10 values were inferior to TNF and IL-6, confirming a Th1 response. 

Studies showed that IFN-γ is a key molecule acting against tumor microenvironments because it can inhibit angiogenesis and cellular proliferation, promote apoptosis in cancer cells, and activate the immune system by antigen presentation [49,50]. In cancers with an HPV presence, IFN-γ is downregulated, and its lower levels are associated with persistent infection and malignancy in cervical lesions [51]. In our study, IFN was inhibited in CD4+, CD8+, and CD56+ lymphocytes in the presence of oncoproteins E6 and E7, and low amounts of IFN were found in the supernatants of lung cancer cells transfected with HPV oncogenes and co-cultured with lymphocytes and monocytes.

Only the transfected cells promoted the increase in the TNF-α levels in this study (Figure 5E and Figure 6E). Other authors showed that the TNF-α cytokine promotes an exacerbated inflammatory response, is present in CIN2/3 lesions, and can stimulate neoangiogenesis and invasive activity of tumor cells [51,52]. 

The cytokine with a higher production observed in this study was IL-6. For both lymphocytes and monocytes cultures, we found a high production of IL-6 in supernatant, especially in HPV E6- and E7-transfected cells (Figure 5 and Figure 6C). The IL-6 cytokine is associated with epithelial–mesenchymal transition, lung cancer development, progression, and the tumor metastasis of non-small cell lung cancer [34]. In the tumor microenvironment, TGF-β, IL-10, and IL-6 suppress NK cell activity and affect immunosuppressive cells and pro-inflammatory cytokines, decreasing the antitumor response of NK cells and promoting subsequent tumor evasion and progression [1].

## 5. Conclusions

It was observed that transfection with E6 and E7 from HPV16 can inhibit pro-inflammatory responses and decrease the activity of cytotoxic and NK cells, leading to lower tumor resistance and enhanced anti-inflammatory responses. These findings confirm that E6 and E7 can facilitate immune evasion and are associated with tumor progression. Additionally, our results reveal an increased expression of E5 in patient samples and antitumor activity in vitro. These findings suggest that E5 could be a promising target for future therapeutic strategies.

## Figures and Tables

**Figure 1 viruses-16-01731-f001:**
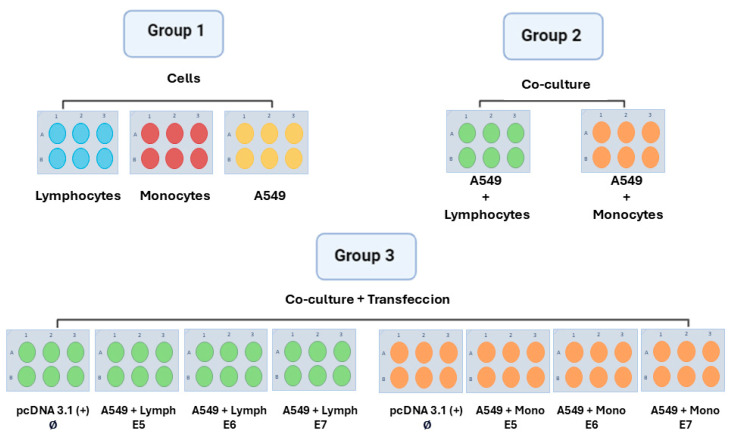
Scheme of the different experimental groups. In group 1, lymphocytes, monocytes, and the A549 lineage were cultured in isolation. In group 2, after the adhesion of tumor cells, lymphocytes, and monocytes were seeded separately on the same plate. In group 3, the A549 tumor line was transfected with the different HPV oncoproteins (E5, E6, and E7) alone, and 24 h later, the lymphocytes and monocytes were added to the culture separately. Five biological replicates were conducted for each group, using lymphocytes and monocytes from five different donors.

**Figure 2 viruses-16-01731-f002:**
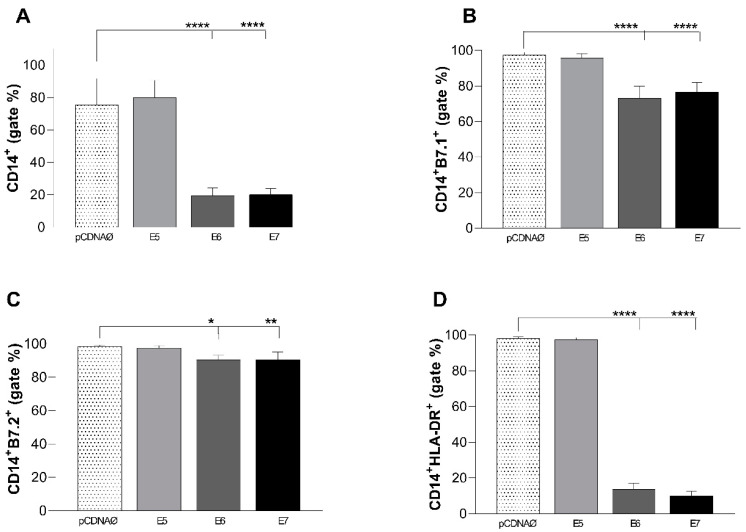
Monocytes CD14+ cultured with the A549 cell lineage (with or without HPV16 oncogenes’ presence). (**A**)—decrease in monocyte amounts. (**B**)—decrease in B7.1 expression. (**C**)—a decrease in B7.2 expression, and (**D**)—a decrease in HLA-DR expression. pcDNAØ corresponds to the A549 linage transfected with the empty vector. The *p* values were obtained by comparing the group MONO+A549 with E5, E6, and E7, respectively. This experiment was conducted with five biological replicates. (* *p* < 0.05, ** *p* < 0.01, **** *p* < 0.0001).

**Figure 3 viruses-16-01731-f003:**
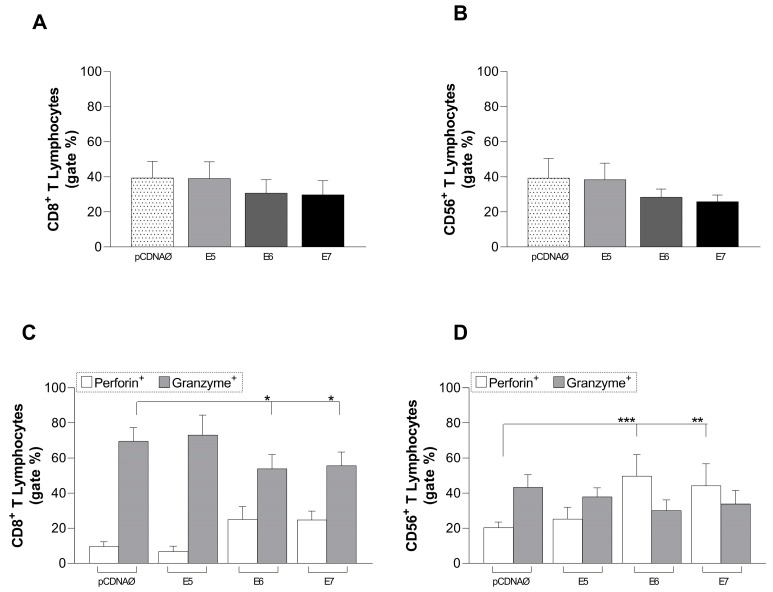
CD8+ and CD56+ lymphocytes’ stimulation promoted by A549 tumor cells transfected with E5, E6, and E7. (**A**)—differential counts of CD8+ T lymphocytes (**B**)—differential count of CD56+ T lymphocytes. (**C**,**D**)—perforin and granzyme intracellular production by CD8+ T and CD56+ lymphocytes, respectively. pcDNAØ corresponds to the A549 linage transfected with the empty vector. * *p* < 0.03; ** *p* < 0.005; *** *p* < 0.0002.

**Figure 4 viruses-16-01731-f004:**
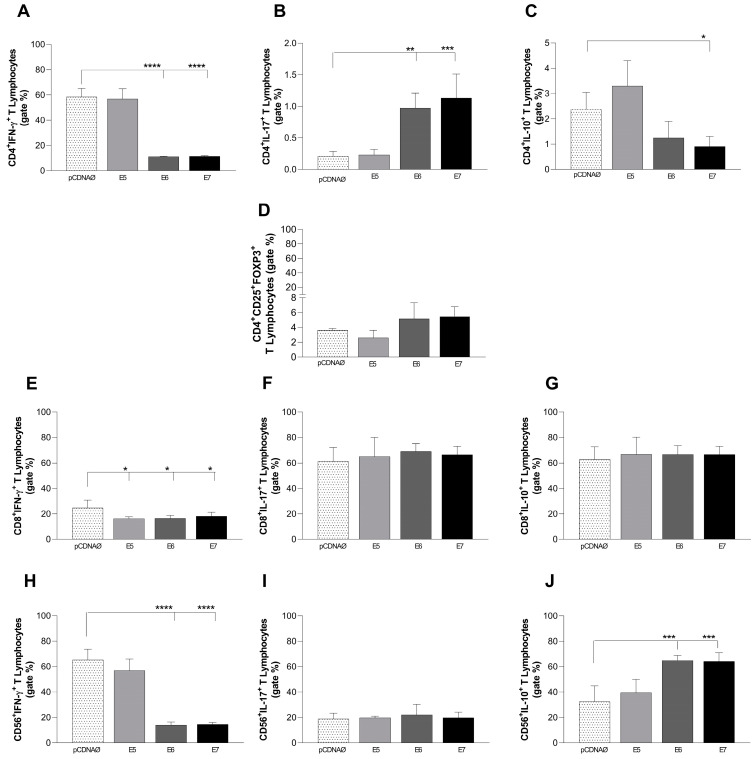
Flow cytometry analysis of intracellular cytokines produced by CD4+, CD8+, and CD56+ lymphocytes against A549 tumor cells. (**A**–**C**)—IFN-γ, IL-10, and IL-17 intracellular production was observed in T CD4+ cells, respectively. (**D**)—CD4 T regulatory cells (CD4+CD25+FOXP3+). (**E**–**G**)—IFN-γ, IL-10, and IL-17 are produced by CD8+ T lymphocytes, respectively. (**H**–**J**)—IFN-γ, IL-10, and IL-17 were produced by CD56+ lymphocytes, respectively. pcDNAØ corresponds to the A549 linage transfected with the empty vector. This experiment was conducted with five biological replicates * *p* = 0.02; ** *p* = 0.001; *** *p* = 0.0003; **** *p* = 0.0001.

**Figure 5 viruses-16-01731-f005:**
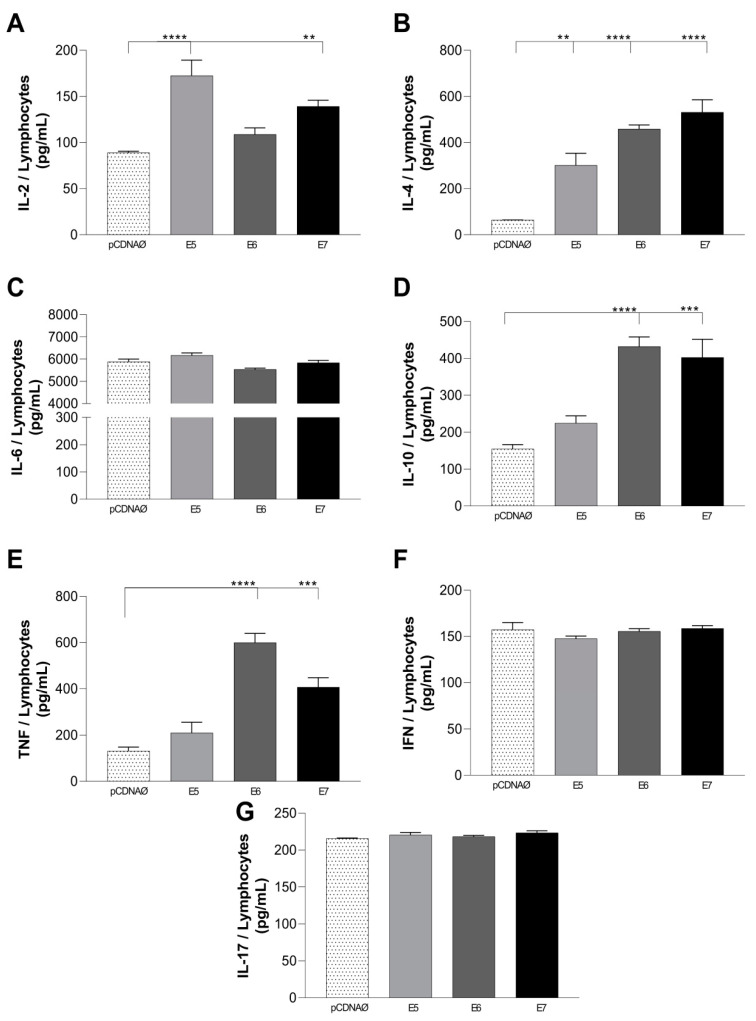
Cytokines present in the supernatant of T lymphocytes cultured with lung tumor cells transfected with oncogenes or an empty vector. (**A**–**G**)—cytokines IL-2, IL-4, IL-6, IL-10, TNF-α, IFN-γ, and IL-17, respectively. pcDNAØ corresponds to the A549 linage transfected with the empty vector. Asterisks represent statistical significance (** *p* < 0.01, *** *p* < 0.001, **** *p* < 0.0001).

**Figure 6 viruses-16-01731-f006:**
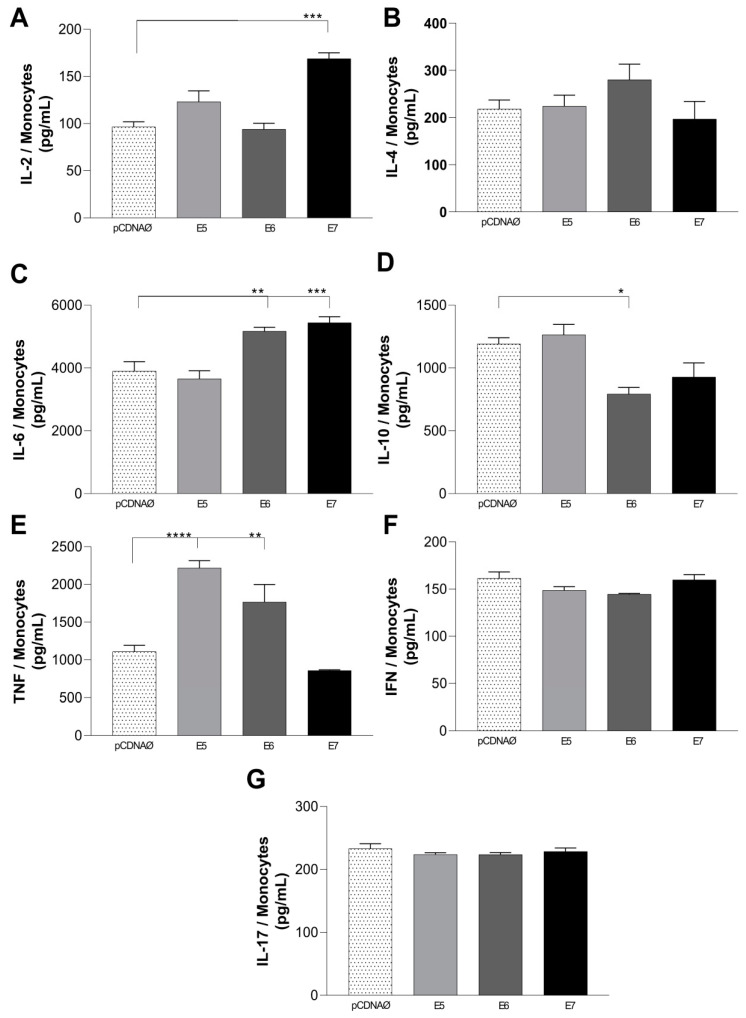
Cytokines present in the supernatant of monocytes co-cultured with lung tumor cells transfected with oncogenes or an empty vector. (**A**–**G**)–cytokines IL-2, IL-4, IL-6, IL-10, TNF-α, IFN-γ, and IL-17, respectively. pcDNAØ corresponds to the A549 linage transfected with the empty vector. Asterisks represent statistical significance (* *p* < 0.05, ** *p* < 0.01, *** *p* < 0.001, **** *p* < 0.0001).

## Data Availability

For access to the information provided in this study, please contact the corresponding author upon request.

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
