# Peer review of "Immune Response Modulation by HPV16 Oncoproteins in Lung Cancer: Insights from Clinical and In Vitro Investigations"

_viruses, 2024, doi:10.3390/v16111731_

Round 1
Reviewer 1 Report
Comments and Suggestions for Authors
In the current study the authors investigated HPV 16 onocoproteins in the context of lung cancer. Both patient samples as well as cell culture (and co-culture) experiments on A549 cells were performed. Furthermore the authors invested significant effort in analysis of many different combinations of cells and transfections with or without co-culture which is highly commendable.
Flow cytometry and bead based interleukin panel assays were used to assess different cells and immunomodulatory components making the study very interesting in the concept.
However, the manuscript is limited by several major problems. The number of patients was quite limited (n=18) and the methods poorly described/presented to the point the patient results do not appear convincing. Flow assays were also suboptimally described and presented. Critically while discussing levels of expression, authors often only present the % gated data instead of actual signal strengths of the markers in a particular population. All comparisons were shown against the empty vector control treated cells while only the supplementary data (poorly crossreferenced in the main text) shows that empty vector itself markedly affects some of the markers making it difficult to interpret the data shown.
While the general issues described above require no particular answer, please provide some response to the detailed comments listed below in a page/line format:
Major
P1 L68 how were patients selected for inclusion? Was general HPV detection and genotyping performed earlier?
P2 L106 the methods imply that “2ΔΔcq“ method was employed, however no comparator (ie normal) samples were presented so it was impossible to calculate delta delta. More details on the calculation process might resolve the uncertainty. For example, were the 2 endogenous genes averaged before calculations, how stable were the 2 endogenous genes, how expressed were they in the samples analysed, how were outliers considered, how were negative results interpreted
P4 L138 it should be specified whether the immunophenotyping was done as single or multicolor experiments (and which combinations). Gating strategy should be shown at least in supplement. What controls were used?
P5 L169 the Figure 2 is unusually presented. Is something at 10exp -8 „expressed“? again this cannot be the output of 2-ddct formula implied above. No quality control of the PCR performed on patient samples is provided so it is impossible to validate the robustness of the results. At least the supplement should contain additional information that strengthens the reliability of the patient data shown.
Currently provided supplementary figure S1 also makes the situation more confusing. The title suggests that the curves present transfection experiments results but the colours are not explained and the results quite inconsistent. For E6 melting curves suggest that at least 2 curves derive from different products than the other major set. In E5 all curves rise above the threshold suggesting that NTC curve (likely ct34) is not completely adequate. Legend lacks a lot of information as well. E7 PCR appears to be poorly performing in cell lines, but in patient samples it was by several orders of magnitude better than E6
For the qPCR analysis of transfected A549 cells where untransfected cells were available the aforementioned “2ΔΔcq“ method could have been performed, however, this data was not shown either in supplement or main text that might give a suggestion on how efficient were the expressions from plasmids in transfected cells. (mock transformed Ct values could be arbitrarily set to 45 for the purpose of ΔΔ calculation
P5 L179 and elsewhere the authors evaluate the expression of different molecules (ie B7.1 IL10…etc) however, the data provided only shows the percentage of cells within undisclosed gates and never directly compares or presents the actual quantity (signal strength) of the targets within a selected population, ie with histograms of selected subpopulations. This problem is especially noticeable at line 215 “increase of intracellular perforin and decrease of intracellular granzyme“ which is factually inconsistent with the data since the data only shows the number of cells positive (with more than an arbitrary level) for either protein instead of intracellular amounts of individual cells. There is of course a correlation between the two concepts but they are not synonymous and authors should clearly distinguish them and provide data visualizing both aspects separately.
P12 L287 the statement that E5 “ enhanced the expression of surface molecules on monocytes responsible for antigen presentation” is not supported by the data shown on figures 3, 4,5 or 6
P12 L 292-294 the sentence is not supported by the data shown. There is no reference to the closest table (Supplementary table 2) but that table also doesn’t clearly support the statement
Possibly the authors should acknowledge that the work was done on a single (adenocarcinoma cell line) infected with HPV16 which usually doesnt lead to adenocarcinomas. HPV18 would be the type leading to adenocarcinomas. Thus the results might not be directly translatable to all lung cancer contexts
Minor
P2 L72 The name of the ethical board should be specified
P2 L86 HPV targets (including HPV types) should be declared in the main manuscript
P2 L86 was the PCR done in duplicate/triplicate, what were the positive/negative controls
P3 L102 the cell quantity is likely erroneous “105 cells/well“ number 5 should be in superscript? Same applies to P3 L123 and P3 L125
P3 L128 apparently all cells were stained for flow cytometry, not only immune cells
P4 L130 Figure 1. Within Group 3 subpanel the pCDNA 3.1(+) is repetitive and could be listed only for “Empty vector”. For modified vectors it might be better to list only the transfected gene (E5, E6, E7) however, the same panel would benefit from clearly identifying the cells studied instead of only relying on color patterns similarity to Group 2. Figure 1 legend might also indicate “number of replicates” at the end
P4 L146 humoral investigation should refer to cytokine investigation
P6 L181 the resolution of Figure 3 is suboptimal. Number of replicates could be stated to make the figure more standalone
P7 L201 the resolution of Figure 4 is suboptimal. Number of replicates could be stated to make the figure more standalone
Figures 4 and 6 could be combined in a way that IFN, IL10 and IL17 are shown for all 3 subsets (CD4, CD8 and CD56) are shown on the same figure. Legend of figure 6 should include information that this was flow cytometry assessment of co-cultures to make it distinct from supernatant bead array experiments
Many Figures use only pcDNA0 group as a negative control. However, it might be worthwhile to show results of Lymphocytes/monocytes alone (Group 1 on Figure 1) in comparison to put the increases/decreases in the context of the situation of cells alone directly in the manuscript instead only in supplement. Current presentation of the results ignores 2/3rds of Figure 1 (no data is shown for Groups 1 or 2) making the study appear less comprehensive.
On the other hand Supplementary figure S4 A/C/D and S6 have completely different interpretations from the same panels shown on Figure 4. Apparently the act of transfection with the empty vector itself changes the lymphocyte patterns greatly in some experiments. Demonstrated best on Figures S7 and S8
Page 10 L 264 section subtitle is erroneous as the section deals with monocytes while the title implies lymphocytes.
Right side of supplementary figure S6 was cropped out
Discrepancies between data of cell flow cytometry and CBA assay should be discussed. For example IFN was not changed in CBA but positive cell subset was consistently found as less abundant in flow cytometry by in all HPV E7 subsets (CD4, CD8 and CD56)
Trivial
Consider rearranging order of references 2 and 3 so that at line 43 you could state [1,2], on line 39 [3-5], and [1,2,18] on line 59,
Comments on the Quality of English Language
P3 L96 Typo “sequency“
P3 L112 typo „E6 e E7“
P4 L148 "All samples that adopted a normal distribution" grammar...
P4 L156 "evaluated samples were male" grammar. It should be "samples were from male patients", or only "evaluated patients were male"
P5 L158 consider rounding all percentages to 1 decimal point and fix the decimal comma typos “83,33%“
P6 L199 "but without statistical values" grammar also at L214
P7 L230 "did not suffer a change in the cytokines production against...challenged against transfected A549 cells different behaviors could be observed" grammar
P10 L263 p value legend uses decimal comma instead of decimal point (and likely elsewhere)
P11 L278 missing fullstop between sentences “vector Asterisks“
P12 L305 typo „samples. [22].“
P13 L342 grammar “which is not favors tumor development“
Author Response
1) P1 L68 how were patients selected for inclusion? Was general HPV detection and genotyping performed earlier?
Answer: All samples received by the Oswaldo Cruz Hospital had their RNA extracted, and only those with adequate quality, as determined by agarose gel analysis and Nanodrop measurements, were used in this study. Regarding the detection of HPV in the patients' samples, this was performed in a previous study published by our group. Additional information has been included in the manuscript for better understanding
de Oliveira, T.H.A., do Amaral, C.M., de França São Marcos, B. et al. Presence and activity of HPV in primary lung cancer. J Cancer Res Clin Oncol 144, 2367–2376 (2018). https://doi.org/10.1007/s00432-018-2748-82)
2) P2 L106 the methods imply that “2ΔΔcq“ method was employed, however no comparator (ie normal) samples were presented so it was impossible to calculate delta delta. More details on the calculation process might resolve the uncertainty. For example, were the 2 endogenous genes averaged before calculations, how stable were the 2 endogenous genes, how expressed were they in the samples analysed, how were outliers considered, how were negative results interpreted
Answer: The 2ΔΔcq calculation was based on two internal control genes (ACTB and EEF1A1), which were properly validated in previously published studies (Zhan et al., 2014). Additionally, relative expression was calculated using the HPV16-positive cell line (C3), which expresses viral oncogenes, as the control for the reactions. Samples were considered negative when Cq values were equal to or greater than 35.
Zhan, C.; Zhang, Y.; Ma, J.; Wang, L.; Jiang, W.; Shi, Y.; Wang, Q. Identification of Reference Genes for QRT-PCR in Human Lung Squamous-Cell Carcinoma by RNA-Seq. Acta Biochim Biophys Sin (Shanghai) 2014, 46, 330–337, doi:10.1093/abbs/gmt153.
3) P4 L138 it should be specified whether the immunophenotyping was done as single or multicolor experiments (and which combinations). Gating strategy should be shown at least in supplement. What controls were used?
Answer: Immunophenotyping was conducted using multicolor experiments, with all assays performed on a flow cytometer (Accuri BD®), capable of detecting up to four fluorescence channels. Different combinations of up to four antibodies were applied per assay to ensure that the fluorescence intensity remained within the detection limits of the equipment.
The A549 cell line, as well as isolated lymphocytes and monocytes, were used as controls. These cells were both stained and unstained to identify the cell populations and establish the baseline fluorescence levels, facilitating the setting of gates. In the supplementary material, the gating strategy for all populations was included.
4) P5 L169 the Figure 2 is unusually presented. Is something at 10exp -8 „expressed“? again this cannot be the output of 2-ddct formula implied above. No quality control of the PCR performed on patient samples is provided so it is impossible to validate the robustness of the results. At least the supplement should contain additional information that strengthens the reliability of the patient data shown.
Answer: The expression graph shown in Figure 2 was represented in log scale due to the low expression of viral oncogenes in lung cancer samples, which has been previously reported in studies showing that lung tumors have a low HPV viral load.
Osorio, J.C.; Candia-Escobar, F.; Corvalán, A.H.; Calaf, G.M.; Aguayo, F. High-Risk Human Papillomavirus Infection in Lung Cancer: Mechanisms and Perspectives. Biology 2022, 11, 1691. https://doi.org/10.3390/biology11121691
5) Currently provided supplementary figure S1 also makes the situation more confusing. The title suggests that the curves present transfection experiments results but the colours are not explained and the results quite inconsistent. For E6 melting curves suggest that at least 2 curves derive from different products than the other major set. In E5 all curves rise above the threshold suggesting that NTC curve (likely ct34) is not completely adequate. Legend lacks a lot of information as well. E7 PCR appears to be poorly performing in cell lines, but in patient samples it was by several orders of magnitude better than E6.
Answer: Figure S1 shows the expression graphs of oncogenes transfected into A549 cells. The figures on the left display the amplification curves of the genes, and the figures on the right show the melting curves. These data were included to confirm the transfection and expression of oncogenes in the in vitro experiments. Each color in the graph represents a biological replicate of each group, which was performed in duplicate, and the average Cq values were subsequently analysed. The HPV16-positive cell line (C3) was used as a control for each gene evaluated. The highest peak observed in the melting curve of the E6 gene corresponds to C3, which has higher expression of this gene. The data regarding E5 provided in the supplementary material were mistakenly replaced with data used during the testing phase with different primers. The correct data have now been added.
6) For the qPCR analysis of transfected A549 cells where untransfected cells were available the aforementioned “2ΔΔcq“ method could have been performed, however, this data was not shown either in supplement or main text that might give a suggestion on how efficient were the expressions from plasmids in transfected cells. (mock transformed Ct values could be arbitrarily set to 45 for the purpose of ΔΔ calculation.
Answer: The 2ΔΔcq calculation for transfected A549 cells was not performed because the A549 cell line does not naturally express HPV16. Therefore, there would be no oncogene expression, making it impossible to calculate relative expression using A549 in comparison to HPV16 oncogene expression.
7) P5 L179 and elsewhere the authors evaluate the expression of different molecules (ie B7.1 IL10…etc) however, the data provided only shows the percentage of cells within undisclosed gates and never directly compares or presents the actual quantity (signal strength) of the targets within a selected population, ie with histograms of selected subpopulations. This problem is especially noticeable at line 215 “increase of intracellular perforin and decrease of intracellular granzyme“ which is factually inconsistent with the data since the data only shows the number of cells positive (with more than an arbitrary level) for either protein instead of intracellular amounts of individual cells. There is of course a correlation between the two concepts but they are not synonymous and authors should clearly distinguish them and provide data visualizing both aspects separately.
Answer: Section 3.3 has been rewritten for improved clarity and to aid reader comprehension, as well as section 3.5. Additionally, the gating strategies have been included in the supplementary material to further assist in understanding the data presented.
8) P12 L287 the statement that E5 “ enhanced the expression of surface molecules on monocytes responsible for antigen presentation” is not supported by the data shown on figures 3, 4,5 or 6.
Answer: The sentence was modified to better represent the results obtained.
9) P12 L 292-294 the sentence is not supported by the data shown. There is no reference to the closest table (Supplementary table 2) but that table also doesn’t clearly support the statement.
Answer: The statistical analysis of the biological samples was not included in the manuscript, as the data were not statistically significant due to the small sample size, as noted in the supplementary data.
10) Possibly the authors should acknowledge that the work was done on a single (adenocarcinoma cell line) infected with HPV16 which usually doesnt lead to adenocarcinomas. HPV18 would be the type leading to adenocarcinomas. Thus the results might not be directly translatable to all lung cancer contexts
Answer: It is known that the evaluation of a single lung cell line transfected with HPV may not be representative of all types of lung cancer. However, lung adenocarcinoma, the histological type of the A549 cell line, represents a significant portion of the histological types associated with HPV16 and HPV18 (Dey Parama, 2024; Jing-Yang-Huang, 2022; Telma Siqueira, 2024; Dania Nachira, 2023)
Dey Parama, Bandari BharathwajChetty, Sujitha Jayaprakash, E. Hui Clarissa Lee, Elina Khatoon, Mohammed S. Alqahtani, Mohamed Abbas, Alan Prem Kumar, Ajaikumar B. Kunnumakkara. The emerging role of human papillomavirus in lung cancer. Life Sciences. Volume 351. 2024. 122785. ISSN 0024-3205. https://doi.org/10.1016/j.lfs.2024.122785.
Huang Jing-Yang , Lin Chuck , Tsai Stella Chin-Shaw , Lin Frank Cheau-Feng. Human Papillomavirus Is Associated With Adenocarcinoma of Lung: A Population-Based Cohort Study.Frontiers in Medicine. Volume 9. 2022. ISSN 2296-858X. http://doi.org/10.3389/fmed.2022.932196
Sequeira, T.; Pinto, R.; Cardoso, C.; Almeida, C.; Aragão, R.; Almodovar, T.; Bicho, M.; Bicho, M.C.; Bárbara, C. HPV and Lung Cancer: A Systematic Review. Cancers 2024, 16, 3325. https://doi.org/10.3390/cancers16193325
Nachira, D.; Congedo, M.T.; D’Argento, E.; Meacci, E.; Evangelista, J.; Sassorossi, C.; Calabrese, G.; Nocera, A.; Kuzmych, K.; Santangelo, R.; et al. The Role of Human Papilloma Virus (HPV) in Primary Lung Cancer Development: State of the Art and Future Perspectives. Life 2024, 14, 110. https://doi.org/10.3390/life14010110
11) P2 L72 The name of the ethical board should be specified
Answer: The name of the ethics committee has been included.
12) P2 L86 HPV targets (including HPV types) should be declared in the main manuscript
Answer: The target has been included in the main manuscript.
13) P2 L86 was the PCR done in duplicate/triplicate, what were the positive/negative controls
Answer: The information has been included in the manuscript
14) P3 L102: The cell quantity is likely erroneous; '10^5 cells/well'—the number 5 should be in superscript. The same applies to P3 L123 and P3 L125.
Answer: The necessary corrections have been made
15) P3 L128 apparently all cells were stained for flow cytometry, not only immune
Answer: The term 'immune' has been removed from the sentence
16) P4 L130 Figure 1. Within Group 3 subpanel the pCDNA 3.1(+) is repetitive and could be listed only for “Empty vector”. For modified vectors it might be better to list only the transfected gene (E5, E6, E7) however, the same panel would benefit from clearly identifying the cells studied instead of only relying on color patterns similarity to Group 2. Figure 1 legend might also indicate “number of replicates” at the end
Answer: The changes to the figure and caption have been made according to the suggestions.
17) P4 L146: 'Humoral investigation' should refer to 'cytokine investigation
Answer: The term 'humoral' has been corrected to 'cytokine.
18) P6 L181 the resolution of Figure 3 is suboptimal. Number of replicates could be stated to make the figure more standalone
Answer: Figure 3 has been enlarged for better resolution. The number of biological replicates has been added to the caption.
19) P7 L201 the resolution of Figure 4 is suboptimal. Number of replicates could be stated to make the figure more standalone
Answer: Figure 4 has been enlarged for better resolution. The number of biological replicates has been added to the caption
20) Figures 4 and 6 could be combined in a way that IFN, IL10 and IL17 are shown for all 3 subsets (CD4, CD8 and CD56) are shown on the same figure. Legend of figure 6 should include information that this was flow cytometry assessment of co-cultures to make it distinct from supernatant bead array experiments
Answer: Figures 4 and 6 have been combined as suggested, and the legend has been modified for better reader comprehension
21) Many Figures use only pcDNA0 group as a negative control. However, it might be worthwhile to show results of Lymphocytes/monocytes alone (Group 1 on Figure 1) in comparison to put the increases/decreases in the context of the situation of cells alone directly in the manuscript instead only in supplement. Current presentation of the results ignores 2/3rds of Figure 1 (no data is shown for Groups 1 or 2) making the study appear less comprehensive. On the other hand Supplementary figure S4 A/C/D and S6 have completely different interpretations from the same panels shown on Figure 4. Apparently the act of transfection with the empty vector itself changes the lymphocyte patterns greatly in some experiments. Demonstrated best on Figures S7 and S8
Answer: We did not include in the main manuscript the information about isolated lymphocytes/monocytes or the co-culture without transfection, as we consider that the empty vector would be the best comparison model with the groups transfected with the oncogenes. This is because transfection alone may trigger an immune response (Montakhab-Yega et al., 2022), making the group with the empty vector sufficient to evaluate the impact of the oncogenes. By doing so, we highlight only the activity of the oncoproteins in the immune response, in line with the objective of the study. If the reader is interested in viewing the results of the other groups, this information is available in the supplementary material
Montakhab-Yeganeh H, Shafiei R, Najm M, Masoori L, Aspatwar A, Badirzadeh A. Immunogenic properties of empty pcDNA3 plasmid against zoonotic cutaneous leishmaniasis in mice. PLoS One. 2022 Feb 15;17(2):e0263993. doi: 10.1371/journal.pone.0263993. PMID: 35167596; PMCID: PMC8846536.
22) Page 10 L 264 section subtitle is erroneous as the section deals with monocytes while the title implies lymphocytes.
Answer: The subtitle has been corrected from 'lymphocytes' to 'monocytes.
23) Right side of supplementary figure S6 was cropped out
Answer: Figure S6 has been corrected as noted by the reviewer.
24) Discrepancies between data of cell flow cytometry and CBA assay should be discussed. For example IFN was not changed in CBA but positive cell subset was consistently found as less abundant in flow cytometry by in all HPV E7 subsets (CD4, CD8 and CD56)
Answer: The IFN results were discussed on P14 starting from line 397, highlighting the decrease of IFN within CD4+, CD8+, and CD56+ cells, as well as a global decrease of IFN in the supernatant, revealed by the low concentration (150 pg/ml) when compared to the other cytokines evaluated in the supernatant.
25) Consider rearranging order of references 2 and 3 so that at line 43 you could state [1,2], on line 39 [3-5], and [1,2,18] on line 59.
Answer: References 2 and 3 have been reorganized as suggested
26) Comments on the Quality of English Language
P3 L96 Typo “sequency“
P3 L112 typo „E6 e E7“
P4 L148 "All samples that adopted a normal distribution" grammar...
P4 L156 "evaluated samples were male" grammar. It should be "samples were from male patients", or only "evaluated patients were male"
P5 L158 consider rounding all percentages to 1 decimal point and fix the decimal comma typos “83,33%“
P6 L199 "but without statistical values" grammar also at L214
P7 L230 "did not suffer a change in the cytokines production against...challenged against transfected A549 cells different behaviors could be observed" grammar
P10 L263 p value legend uses decimal comma instead of decimal point (and likely elsewhere)
P11 L278 missing fullstop between sentences “vector Asterisks“
P12 L305 typo „samples. [22].“
P13 L342 grammar “which is not favors tumor development“
Answer: All previously mentioned points have been corrected.
Please do not hesitate to contact me if further information is needed.
We appreciate your insightful comments, which have helped improve the clarity and precision of our work.
Sincerely,
ANTONIO CARLOS DE FREITAS, PH.D
Associate Professor
Head of Laboratory of Molecular Studies and Experimental Therapy (LEMTE)
Department of Genetics
Federal University of Pernambuco
Recife, Pernambuco -Brazil

Reviewer 2 Report
Comments and Suggestions for Authors
The manuscript is precise and well-written. Certain minor mistakes can be corrected. For instance, the subtitles of 3.7 and 3.8 are the same. What is the test for HPV presence in Table S2? It needs to be mentioned. The format of reference is not consistent.
Author Response
Reviewer 2
1) The manuscript is precise and well-written. Certain minor mistakes can be corrected. For instance, the subtitles of 3.7 and 3.8 are the same. What is the test for HPV presence in Table S2? It needs to be mentioned. The format of reference is not consistent.
Answer: Thank you for your valuable feedback and for highlighting these important details. We have carefully addressed the points you raised. The subtitles of Sections 3.7 and 3.8 have been corrected, and we have included the test for HPV presence in Table S2 as suggested. Additionally, the format of the references has been revised to ensure consistency throughout the manuscript.
Please do not hesitate to contact me if further information is needed.
We appreciate your insightful comments, which have helped improve the clarity and precision of our work.
Sincerely,
ANTONIO CARLOS DE FREITAS, PH.D
Associate Professor
Head of Laboratory of Molecular Studies and Experimental Therapy (LEMTE)
Department of Genetics
Federal University of Pernambuco
Recife, Pernambuco -Brazil
Round 2
Reviewer 1 Report
Comments and Suggestions for Authors
The revised manuscript by de França São Marcos et al addresses most of the technical issues with the original but did not address some of the major problems or introduced other issues.
Regarding the sample inclusion criteria question the authors added a reference to their previous study [19 – Marcos Microbiology, 2022]. Neither the new additions or the reference explains how this particular 18 cases were selected for inclusion. It is unlikely that pulmonology department of Oswaldo Cruz University Hospital treated a total of 18 lung cancer patients throughout these years. Furthermore now the authors state „All HPV16-positive biological samples were selected based on the quality of RNA 75 extraction. (P2L75)“, but this implies that the study population is biased towards HPV positive lung cancer cases. Indeed the authors find 83% of HPV positive cases (p5L166) which is higher than 2021 meta analysis showing an average of 13.5% HPV positivity in lung cancer (https://pmc.ncbi.nlm.nih.gov/articles/PMC8388180/) or the most recent showing 0-69% prevalence (https://www.mdpi.com/2072-6694/16/19/3325). The clear stating of the inclusion criteria might explain WHY those 18 patients had so high HPV positivity beyond what is expected?
Another critical problem arising from the revision is the fact apparently the authors already published the E5, E6 and E7 qPCR results from the same cohort previously. The reference 19 includes the sentence “On the other hand, we analyzed the expression of E5, E6, and E7 oncogenes from all FF samples (19 samples). 10/19 were positive for E5, 2/19 were positive for E6 oncogene, E7 was detected in 4/19 samples, and all four were previously negative for HPV by the conventional PCR results. The only oncogenes expressed were from HPV 16.” Thus the current manuscript should be revised to acknowledge that only E2 qPCR suggested in the materials section was newly done with the rest being done earlier.
There is also a mismatch in number of cases analysed in the current and previous study (18 vs 19) as well as HPV positivity reported in the two studies on the same cases. In their previous study [19] the authors described 19 fresh frozen biopsies from lung cancer patients that were also assessed by consensus HPV specific PCR (and sequencing for genotyping). In that cohort there were only 4/19 HPV16 positive cases “The 19 FF samples were screened separately, and five samples were positive for hrHPV on the first screening, 80% positive for HPV 16, and 20% for HPV 18.”. Furthermore the authors added the E6/E7 qPCR cases but still the total was less. “oncogenes were expressed in 14/19, 13/14 positive for HPV 16, and 1/14 for HPV 18. Because of that, all the subsequent analysis was performed assuming 14/19 samples as HPV+.”. In the current study there were 15 out of 18 HPV positive cases (supplementary table 2). Also previously, 4/19 were considered positive for HPV16 E7 but new Figure 2 clearly shows 6 very similar cases positive for E7. Previously 10/19 were considered positive but now the figure 2 shows at least 13 individual cases „positive“ for E5?
Either additional paraffin embedded material was used from the initial study (with HPV16+ positivity bias), or the current methods section does not correspond to what was actually done for the current study.
Another major issue was the use of ΔΔct method. The authors replied that „relative expression was calculated using the HPV16-positive cell line (C3), which expresses viral oncogenes“
This further complicates the manuscript methods since the C3 line is undeclared/unreferenced. Brief search identifies C3 line as mouse embryonal HPV transformed cell line (https://www.cellosaurus.org/CVCL_DC63). While using an unrelated reference is technically possible in ddCt method (https://www.sciencedirect.com/science/article/pii/S1046202301912629), the manuscript should clearly indicate that the fold changes are relative to C3 cell line. For example the sentence “E5 was overexpressed in most lung tumors“ (P5L162) is very problematic since the comparator was mouse embryonal cell line C3 and not healthy lung tissue. Overexpression versus normal is only implied but was not actually calculated. And E5 was found to be underexpressed versus C3 line. Also, the current manuscript should well explain why the qPCR data already published within reference 19 is shown again in the current manuscript. Not declaring this explicitly might lead the same data being included multiple times in subsequent meta analyses and skewing their results unjustifiably and excessively towards unusually (and possibly biased) high HPV prevalence in lung cancer.
P2 L86 “hpv16” should be “HPV”
P3 L 119 “The 2ΔΔcq was calculated to determine the specific gene expression (E5, E6 and E7)“ lacks mentioning E2
P3 L128 “105 cells” should be 10exp5
P5 L166 “ 83,33% of the“ the use of decimal comma was not fixed, and this is likely overlapping with the previous manuscript
P12 L311 „To date, only one study conducted by our group evaluated the expression of E5 in 311 HPV-positive lung cancer samples, which found that the oncogene E5 was the most prevalent and highly expressed among the samples. [19].“ the sentence is problematic if indeed the same cases are reanalyzed in the current study
The other changes made to the manuscript are appreciated and the effort is acknowledged
Author Response
Reviewer 1 – Round 2
- Regarding the sample inclusion criteria question the authors added a reference to their previous study [19 – Marcos Microbiology, 2022]. Neither the new additions or the reference explains how this particular 18 cases were selected for inclusion. It is unlikely that pulmonology department of Oswaldo Cruz University Hospital treated a total of 18 lung cancer patients throughout these years. Furthermore now the authors state „All HPV16-positive biological samples were selected based on the quality of RNA 75 extraction. (P2L75)“, but this implies that the study population is biased towards HPV positive lung cancer cases. Indeed the authors find 83% of HPV positive cases (p5L166) which is higher than 2021 meta analysis showing an average of 13.5% HPV positivity in lung cancer (https://pmc.ncbi.nlm.nih.gov/articles/PMC8388180/) or the most recent showing 0-69% prevalence (https://www.mdpi.com/2072-6694/16/19/3325). The clear stating of the inclusion criteria might explain WHY those 18 patients had so high HPV positivity beyond what is expected?
Answer:
The 18 samples analyzed in this study were obtained from a sample bank belonging to our research group. These are remaining samples collected between 2015 and 2018. The number of samples is limited due to the difficulty of obtaining this type of material, considering the highly invasive nature of the required surgeries and the challenges in accessing patient information from state public hospitals. Samples were randomly selected from tumor material, without specification of tumor subtype or the sociodemographic characteristics of the patients. This procedure minimizes bias and characterizes a study with a blinded sampling approach, promoting an impartial data analysis.
The high frequency of HPV positivity in the samples analyzed in our study reflects the fact that this region has a high incidence of HPV, especially in cervical cancers (Colpani et al., 2020). According to Sung et al., 2021, cervical cancer is the second most common cancer in countries with low and medium Human Development Index (HDI). Furthermore, the prevalence of HPV infection in the cervix can vary significantly within the country, which, due to its continental size, has distinct sociodemographic characteristics. In Brazil, the Northeast is the second region with the highest incidence (17.59 per 100,000 inhabitants) (INCA, 2022). Studies in other countries also indicate a high variability in the incidence of HPV-associated lung cancers (Tsyganov et al., 2019).
Colpani V, Soares Falcetta F, Bacelo Bidinotto A, Kops NL, Falavigna M, Serpa Hammes L, Schwartz Benzaken A, Kalume Maranhão AG, Domingues CMAS, Wendland EM. Prevalence of human papillomavirus (HPV) in Brazil: A systematic review and meta-analysis. PLoS One. 2020 Feb 21;15(2):e0229154. doi: 10.1371/journal.pone.0229154. PMID: 32084177; PMCID: PMC7034815.
INCA (2022) Estimativa 2023 Incidância de Câncer no Brasil. ISBN 978-65-88517-10-9
Hyuna Sung, Jacques Ferlay, Rebecca L. Siegel, Mathieu Laversanne, Isabelle Soerjomataram, Ahmedin Jemal, Freddie Bray. Global Cancer Statistics 2020: GLOBOCAN Estimates of Incidence and Mortality Worldwide for 36 Cancers in 185 Countries. (2021). CA: A Cancer Journal for Clinicians. doi: 10.3322/caac.21660
Tsyganov MM, Pevzner · A M, Ibragimova · M K, Deryusheva · I V and Litviakov · N V (2019) Human papillomavirus and lung cancer: an overview and a meta-analysis. J Cancer Res Clin Oncol 145:1919–1937. doi: 10.1007/s00432-019-02960-w
- Another critical problem arising from the revision is the fact apparently the authors already published the E5, E6 and E7 qPCR results from the same cohort previously. The reference 19 includes the sentence “On the other hand, we analyzed the expression of E5, E6, and E7 oncogenes from all FF samples (19 samples). 10/19 were positive for E5, 2/19 were positive for E6 oncogene, E7 was detected in 4/19 samples, and all four were previously negative for HPV by the conventional PCR results. The only oncogenes expressed were from HPV 16.” Thus the current manuscript should be revised to acknowledge that only E2 qPCR suggested in the materials section was newly done with the rest being done earlier.
Answer: We have acknowledged your suggestions and revised the manuscript accordingly to clarify that only the qPCR analysis for the E2 gene was newly conducted in this study, while the qPCR results for E5, E6, and E7 were previously published with the same cohort, as referenced. However, a reanalysis of the oncoproteins was conducted and compared with previous analyses to validate the results of E2.
- There is also a mismatch in number of cases analysed in the current and previous study (18 vs 19) as well as HPV positivity reported in the two studies on the same cases. In their previous study [19] the authors described 19 fresh frozen biopsies from lung cancer patients that were also assessed by consensus HPV specific PCR (and sequencing for genotyping). In that cohort there were only 4/19 HPV16 positive cases “The 19 FF samples were screened separately, and five samples were positive for hrHPV on the first screening, 80% positive for HPV 16, and 20% for HPV 18.”. Furthermore the authors added the E6/E7 qPCR cases but still the total was less. “oncogenes were expressed in 14/19, 13/14 positive for HPV 16, and 1/14 for HPV 18. Because of that, all the subsequent analysis was performed assuming 14/19 samples as HPV+.”. In the current study there were 15 out of 18 HPV positive cases (supplementary table 2). Also previously, 4/19 were considered positive for HPV16 E7 but new Figure 2 clearly shows 6 very similar cases positive for E7. Previously 10/19 were considered positive but now the figure 2 shows at least 13 individual cases „positive“ for E5?
Either additional paraffin embedded material was used from the initial study (with HPV16+ positivity bias), or the current methods section does not correspond to what was actually done for the current study.
Answer: Indeed, the previous cited study analyzed 19 FF samples, of which 13 were HPV16 positive and only 1 was HPV18 positive. In the current study, the HPV18 sample was not included, as the objective was to evaluate only HPV16 activity in lung cancers, resulting in 18 lung tumor samples, 13 of which were positive for one of the HPV16 genes. There was an error in counting the positive proteins in our current study, which has now been corrected.
- Another major issue was the use of ΔΔct method. The authors replied that „relative expression was calculated using the HPV16-positive cell line (C3), which expresses viral oncogenes“
This further complicates the manuscript methods since the C3 line is undeclared/unreferenced. Brief search identifies C3 line as mouse embryonal HPV transformed cell line (https://www.cellosaurus.org/CVCL_DC63). While using an unrelated reference is technically possible in ddCt method (https://www.sciencedirect.com/science/article/pii/S1046202301912629), the manuscript should clearly indicate that the fold changes are relative to C3 cell line. For example the sentence “E5 was overexpressed in most lung tumors“ (P5L162) is very problematic since the comparator was mouse embryonal cell line C3 and not healthy lung tissue. Overexpression versus normal is only implied but was not actually calculated. And E5 was found to be underexpressed versus C3 line. Also, the current manuscript should well explain why the qPCR data already published within reference 19 is shown again in the current manuscript. Not declaring this explicitly might lead the same data being included multiple times in subsequent meta analyses and skewing their results unjustifiably and excessively towards unusually (and possibly biased) high HPV prevalence in lung cancer.
Answer: We have recalculated the 2ΔΔCt values by removing the C3 cell line and instead using a positive HPV16 cervical tumor sample. The graph has been redone and relocated to the supplementary material. Unfortunately, we cannot compare with healthy lung tissue, as we do not possess non-tumor samples that are positive for HPV, making it impossible to calculate the relative expression of the viral genes. Furthermore, the information regarding the qPCR data already published in the previous study was included in the manuscript for better clarity and understanding.
- P2 L86 “hpv16” should be “HPV”
Answer: the term “hpv16” has been corrected to “HPV16”
- P3 L 119 “The 2ΔΔcq was calculated to determine the specific gene expression (E5, E6 and E7)“ lacks mentioning E2
Answer: The expression of E2 was not evaluated for the A549 transfected cell lines; only E5, E6, and E7 were analyzed because the cells were transfected solely with the oncogenes, not with E2. Since the objective of this experiment was to demonstrate that the vector was expressing the oncogenes, a relative expression calculation was not performed. For this reason, this section has been removed from the manuscript.
- P3 L128 “105 cells” should be 10exp5
Answer: “105 cell” has been corrected to “105”
- P5 L166 “ 83,33% of the“ the use of decimal comma was not fixed, and this is likely overlapping with the previous manuscript
Answer: The information has been corrected in manuscript
- P12 L311 „To date, only one study conducted by our group evaluated the expression of E5 in 311 HPV-positive lung cancer samples, which found that the oncogene E5 was the most prevalent and highly expressed among the samples. [19].“ the sentence is problematic if indeed the same cases are reanalyzed in the current study
Answer: We chose not to remove this information, as we had clarified in the manuscript that the expression levels of the oncogenes (E5, E6, and E7) were based on findings from a previously referenced study, with only the expression level of E2 being specifically evaluated in this manuscript.
The other changes made to the manuscript are appreciated and the effort is acknowledged
Thank you for your acknowledgment of the changes made to the manuscript. We appreciate your feedback and are available for any further questions or assistance you may need.
Sincerely,
ANTONIO CARLOS DE FREITAS, PH.D
Associate Professor
Head of Laboratory of Molecular Studies and Experimental Therapy (LEMTE)
Department of Genetics
Federal University of Pernambuco
Recife, Pernambuco -Brazil
